# Associations between the Number of Children, Depressive Symptoms, and Cognition in Middle-Aged and Older Adults: Evidence from the China Health and Retirement Longitudinal Study

**DOI:** 10.3390/healthcare12191928

**Published:** 2024-09-26

**Authors:** Yongze Zhao, Huaxin Mai, Ying Bian

**Affiliations:** 1Department of Public Health and Medicinal Administration, Faculty of Health Sciences, University of Macau, Taipa, Macau, China; 2Unit of Psychiatry, Department of Public Health and Medicinal Administration & Institute of Translational Medicine, Faculty of Health Sciences, University of Macau, Taipa, Macau, China; 3Institute of Chinese Medical Sciences, University of Macau, Taipa, Macau, China; 4State Key Laboratory of Quality Research in Chinese Medicine, University of Macau, Taipa, Macau, China

**Keywords:** family size, depressive symptoms, cognition, middle aged, aged, mediation analysis, CHARLS

## Abstract

*Background*: China’s rapidly aging population presents challenges for cognitive health and mental well-being among the older adults. This study examines how the number of children affects cognitive function in middle-aged and older adults and whether depressive symptoms mediate this relationship. *Methods*: This study analyzed data from waves 1 to 5 (2011–2020) of the China Health and Retirement Longitudinal Study (CHARLS), involving 5932 participants aged 45 and older. Participants were grouped by the number of children: childless, only child and multiple children. We used Logarithmic Generalized Linear Models (LGLMs) to explore the relationships among the number of children, depressive symptoms, and cognitive function. Indirect effect coefficients and 95% bias-corrected and accelerated confidence intervals (BCaCI) were estimated using Simultaneous Equation Models (SEM) with three-stage least squares (3SLS) and the bootstrap method to assess the mediating effect of depressive symptoms. *Results*: In middle-aged and older adults, a negative association was observed between the number of children and overall cognitive functioning (all *p* < 0.01). This association remained significant even after adjusting for covariates in groups with three (*β* = −0.023, *p* < 0.05) and four or more children (*β* = −0.043, *p* < 0.001). Conversely, the positive association between the number of children and depression also persisted after adjusting for covariates, although it weakened as the number of children increased (all *p* < 0.01). Depressive symptoms consistently correlated negatively with overall cognitive function (*p* < 0.001) and partially mediated the relationship between the number of children and cognitive function (pMe = 20.36%, *p* < 0.05). The proportion of the mediating effect attributed to depression was more pronounced in middle-aged and older adults who had experienced the loss of children (pMe = 24.31%) or had two children (pMe = 25.39%), with stronger mediating effects observed in males (pMe = 48.84%) and urban residents (pMe = 64.58%). *Conclusions*: The findings indicate that depressive symptoms partially mediate the relationship between the number of children and cognitive function in middle-aged and older adults in China. These results highlight the significance of considering mental health factors when studying cognitive function in this demographic. Notably, in families without children and those with two children, depressive symptoms play a crucial role in explaining the decline in cognitive function.

## 1. Introduction

The global trend of population aging presents significant challenges, particularly in China, where this demographic transformation is occurring at an unprecedented rate. According to the United Nations and China’s Population and Development Research Centre, China’s older population will reach a peak between 2027 and 2038, with an average annual increase of 11.2 million, ultimately reaching 399 million by 2049 (accounting for 28.9% of the overall population) [1]. The implications of this demographic shift are profound, imparting social, economic, and healthcare systems on a massive scale [2]. One of the most prevalent and damaging age-related conditions is cognitive decline [3]. This decline often leads to diseases such as dementia and Alzheimer’s disease [4], which impose uncertain significant financial burdens on individuals and healthcare systems alike. The mechanisms involved are that aging naturally induces alterations in blood vessels, synaptic plasticity, and neurodegenerative processes, all of which contribute to the decline in cognitive function [3,5,6,7]. Other age-related risk factors, such as chronic illnesses, reduced physical activity, and social isolation, also further exacerbate cognitive decline [8,9,10]. Therefore, understanding the factors that influence cognitive health in older people is critical to addressing the challenges posed by an aging society.

In an effort to control population growth, China implemented the one-child policy in 1979. This policy, which restricts most families to a single child, aimed to slow the rate of population increase and promote economic development. Initially, the one-child policy was successful in reducing China’s total fertility rate from 5.8 in the 1970s to below 2.1 in the 1990s. However, despite its effectiveness in managing population size, the policy has led to numerous social and economic problems, including an aging population, gender imbalance, and psychological and social pressures on families with only one child. Over time, the one-child policy has significantly influenced family structure and social dynamics, particularly in an aging society where the challenges faced by one-child families have become increasingly pronounced. Following the complete implementation of the two-child policy in 2015, the repercussions of the one-child policy persist. Although the new policy allows families to have two children, many still choose to have only one due to economic constraints and high childcare expenses. The long-term effects of the one-child policy warrant further investigation and scholarly attention.

Previous studies have demonstrated that family structure significantly impacts cognitive functioning. For example, family structure during childhood affects early brain development and cognitive performance over their lifespan, and growing up in multigenerational households is linked to higher levels of cognitive functioning in later life [11]. For older adults, a Chinese study revealed that living in a traditional family structure and providing care or assistance with household duties for children enhances cognitive abilities in individuals over 80 years old [12]. Recently, the number of children, a key factor in family structure, has gained attention for its potential impact on cognitive functioning in older adults. Initial attention focused on childless older adults, with a 20-country European study suggesting that non-traditional family forms, such as single and childless couples, may be more vulnerable to social isolation and cognitive decline [13]. Having children is generally associated with better cognitive performance [14]. However, having more children does not necessarily translate to better cognitive function. A Mexican population study found a non-linear relationship between fertility and cognitive ability, with both low and high numbers of children associated with poor cognitive performance [15]. The protective factors related to parenting and cognitive function also vary by gender, generally less beneficial for women [14]. This difference may be related to traditional gender roles, where women who raise more children may have lower levels of social engagement and are more likely to experience depressive symptoms [16].

Other research shows that family structure plays a significant role in mental health and the risk of depression throughout an individual’s life. Young adults raised in single-parent households exhibit greater rates of depressive symptoms [17]. Older adults living alone have a 33% higher chance of experiencing depressive symptoms on average [18]. Furthermore, depression has been linked to cognitive decline in older adults. The mechanisms by which depression induces cognitive impairment in older adults are varied, including small vessel disease, endothelial dysfunction, inflammation, hippocampal atrophy, and deficits of nerve growth factors [19,20]. It has also been shown that older adults with late-life depression (LLD) experience an obvious decline in cognitive function over a short period, with cognitive dysfunction persisting for at least four years after clinical recovery from LLD [21,22].

These findings suggest that depression may mediate the relationship between family structure (e.g., the number of children) and cognitive function in older adults. This study aims to explore whether the number of children affects cognitive functioning in Chinese older adults and whether depression serves as an important mediator in this relationship. To achieve this, we utilize longitudinal data from the China Health and Retirement Longitudinal Study (CHARLS), spanning five waves from 2011 to 2020. The use of longitudinal data allows for more reliable causal inferences than cross-sectional data. Moreover, this study is among the first to systematically examine the mediating role of depression in the relationship between family structure and cognitive function within the Chinese context. The findings have the potential to significantly contribute to the literature on aging and mental health, offering evidence that could inform policy interventions to enhance the cognitive health of older adults in China.

## 2. Materials and Methods

### 2.1. Study Population

This study used the publicly accessible data generated by the National School of Development of Peking University from the China Health and Retirement Longitudinal Study (CHARLS), which is an ongoing longitudinal investigation that focuses on individuals aged 45 and older in China (including 150 counties/districts and 450 villages/urban communities in 28 provinces) [23]. The study sample comprised households with members and employed a multi-stage stratified probability-proportional-to-size (PPS) sampling technique. PPS sampling is a form of probability sampling in which the source of the sample is selected randomly. CHARLS utilizes international data collection methods to ensure comparability, referencing studies such as the Health and Retirement Study (HRS) in the United States. The national baseline survey was conducted in 2011 (wave 1), followed by follow-up surveys using standardized questionnaires in 2013, 2015, 2018, and 2020 (waves 2–5).

The follow-up period for the CHARLS data used in this study was from July 2011 to June 2020. The number of children in the participant cohort was categorized into three groups based on exposure factors: (1) lost; (2) only one child; and (3) two or more children. Cognitive functioning was analyzed as the primary outcome variable, while depressive symptoms were examined as a mediating variable. The baseline cohort population (wave 1) adhered to the following exclusion criteria: (1) individuals aged less than 45 years (n = 448); (2) presence of depressive symptoms (n = 5683); (3) diagnosis of a mental disorder other than depression, such as schizophrenia, anxiety disorders, or bipolar disorder (n = 54); (4) presence of memory-related diseases, including Alzheimer’s disease, dementia, Huntington’s disease, or brain atrophy (n = 89); and (5) history of stroke (n = 166). Participants who were lost to follow-up or who died between waves 2 and 5 (n = 2198) were also excluded from the study. Ultimately, a total of 4822 cohort samples were included in cross-sectional study data (wave 5). After removing missing values (n = 2369), 4808 samples were analyzed for mediated effects. Figure 1 provides a comprehensive overview of the sample processing flow.

### 2.2. Cognitive Function

The assessment of cognitive function across waves 1 to 5 of CHARLS is modeled after the methodology used in the Health and Retirement Study (HRS) [24]. Cognitive function is measured from two dimensions: episodic memory and mental status [25]. The first dimension, episodic memory, involves scoring the respondent’s ability to recall information both immediately and after a delay. Participants were asked to memorize and later recall a list of words presented during the interview, which assesses both short-term and long-term memory performance. The second dimension, mental status, is assessed using the Telephone Interview Cognitive State (TICS) questions, designed to capture an individual’s overall cognitive health. This assessment encompasses general cognitive functions, such as orientation (e.g., knowing the current date, day of the week, and season), calculation abilities (e.g., serial subtraction tasks), and visuospatial skills (e.g., drawing overlapping geometric figures) [26,27]. In this study, total cognitive scores were the sum of the two dimensions and range from 0 to 21, with episodic memory scores ranging from 0 to 10 and mental state scores ranging from 0 to 11. Higher scores indicated better cognitive function, but these scores did not identify specific cognitive impairments.

### 2.3. Depressive Symptoms

The CHARLS survey utilizes the 10-item Centre for Epidemiologic Studies Depression Scale (CES-D10) to evaluate depressive symptoms [28]. The CES-D10 has been validated in Chinese older adults, demonstrating high reliability and validity [29]. The CES-D10 comprises 10 questions designed to assess depressive symptoms experienced by the participants over the past week. Each question offers 4 response options that indicate the frequency of these symptoms during the past week [30]. The total score ranges from 0 to 30, with a cutoff score of  ≥10 points used to identify individuals with significant depressive symptoms among Chinese adults aged 45 and over [26].

### 2.4. Number of Children

Participants were categorized into three groups based on the number of children: childless (0 children), only child (1 child), and multiple children (2 children, 3 children, or 4 or more children). The childless group included individuals who never had children or who lost their only child. The one-child policy has been in effect in China for decades, resulting in a substantial number of one-child families. According to the China Health Statistics Yearbook, there are currently approximately 218 million one-child children in China. The only-child group reflected the impact of China’s “one-child policy,” which led to a significant proportion of one-child families, making this group highly representative of the CHARLS population aged 45 and above [31]. Therefore, we used the one-child status as the reference for this variable.

### 2.5. Covariates

Informed by prior research [32,33], this study selected age, gender, marital status, education level, and type of residence as demographic variables. Additionally, we incorporated 12 risk factors (sleep duration, life satisfaction, smoking status, drinking status, hypertension, diabetes, lung disease, stroke, mental disorder, heart disease, liver disease, and kidney disease) for cognitive impairment, as identified in the Lancet Commission report [34]. Demographic variables included age (calculated by year of birth), gender, residence (urban/rural), education level (below primary school, primary school, middle school, and high school and above), and marital status (single, married). Sleep duration is categorized into three groups based on hours: insufficient sleep (less than or equal to 6 h), sufficient sleep (more than 6 h but less than or equal to 8 h), and excessive sleep (more than 8 h). Life satisfaction is classified into five levels: very good, good, fair, poor, and very poor. Smoking and drinking behaviors were self-reported by participants based on their personal experiences (reference: being never drinking and never smoking). Hypertension, diabetes, lung disease, stroke, mental disorder, heart disease, liver disease, and kidney disease were defined as having been diagnosed by medical professionals. Details can be found in Appendix A.

### 2.6. Statistical Analysis

All statistical analyses were conducted using Stata 17.0 software (StataCorp LLC, College Station, TX, USA). First, descriptive statistics were used to summarize the baseline characteristics of the study participants. Continuous variables were expressed as mean ± standard deviation (SD), and categorical variables were presented as frequencies and percentages. Group comparisons (no children, only child, and two or more children) were performed using t-tests and chi-square tests.

To investigate the relationships among the number of children, depressive symptoms, and cognitive function, we employed Logarithmic Generalized Linear Models (LGLMs). The LGLM allows us to evaluate the nonlinear effects of independent variables on the dependent variable while considering covariates. The logarithmic transformation addresses the non-normal distribution of the dependent variable and enhances the robustness of the model’s fit. The parameters in the model were estimated using the Maximum Likelihood Estimation (MLE) method. The specific formulation of the model used in this study is as follows:(1)$$\log\left(E\left[Y_{i}\right]\right)=\beta_{0}+\beta_{1i}N+\beta_{2i}D+\cdots +\beta_{ki}X_{ki}+\epsilon$$
where $E\left[Y_{i}\right]$ represents the scores for episodic memory, mental status, and total cognitive functioning, respectively. *N* denotes the number of children, *D* indicates depressive symptoms, and $X_{ki}$ refers to a covariate. The models were adjusted for potential confounders, including age, gender, marital status, education level, and type of residence. The parameter $\beta_{ki}$ is to be estimated, and ϵ represents the error terms.

We used Simultaneous Equation Models (SEMs) with three-stage least squares (3SLS) to investigate the mediating effect of depressive symptoms on the relationship between the number of children and cognitive function. SEM allows for the analysis of complex relationships involving multiple dependent variables and mediators. The SEM can be represented by the following system of equations:The equation for the mediator (depressive symptoms) is as follows:(2)$$M_{i}=\alpha_{0}+\alpha_{1}X_{i}+\epsilon_{1i}$$The equation for the outcome (cognitive function) is as follows:
(3)$$Y_{i}=\gamma_{0}+\gamma_{1}X_{i}+\gamma_{2}M_{i}+\epsilon_{2i}$$
where $M_{i}$ represents the mediator (depressive symptoms); $Y_{i}$ represents the outcome (cognitive function); $X_{i}$ represents the independent variable (number of children); $\alpha_{0}$, $\alpha_{1}$, $\gamma_{0}$, $\gamma_{1}$, and $\gamma_{2}$ are the coefficients; and $\epsilon_{1i}$ are the error terms.

In the three-stage least squares (3SLS) method, residuals are initially obtained by estimating each equation using Ordinary Least Squares (OLS). These residuals are then utilized to construct the covariance matrix among the equations. Finally, all equations are estimated jointly using Generalized Least Squares (GLS) to achieve more accurate parameter estimates [35,36]. This approach not only effectively identifies mediating effects but also addresses potential endogeneity issues, thereby enhancing the explanatory and predictive power of the model. When conducting mediation effect tests, the bootstrap method offers significant advantages over traditional methods, such as the Sobel test [37,38]. First, the bootstrap method estimates the distribution of indirect effects through repeated sampling (typically between 1000 and 10,000 iterations), which generates more precise confidence intervals [38]. This method does not rely on the assumption of normality in the sample distribution, making it more robust when handling non-normally distributed data [39].

Sensitivity analyses were performed using Multiple Linear Regression Models to confirm the robustness of our findings. Subgroup analyses were conducted based on gender and place of residence to explore potential differences in the associations. All statistical tests were two-sided, and a *p*-value of less than 0.05 was considered statistically significant.

## 3. Results

### 3.1. Participant Characteristics

The baseline characteristics of participants (N = 5932) with a mean age of 57.07 ± 8.26 years are provided in Table 1. Gender distribution was 53.76% male (n = 3189) and 46.24% female (n = 2743). Regarding residence, 57.42% (n = 3406) lived in rural areas and 42.58% (n = 2526) in urban areas. Educational attainment showed that 33.54% had less than elementary education, 22.95% completed elementary school, 26.74% had secondary education, and 16.78% had high school education or higher. Marital status revealed that 92.85% were married or cohabiting, while 7.15% were divorced, separated, widowed, or never married.

Participants had a mean total cognitive score of 12.71, an episodic memory score of 3.96 (SD = 1.65), and a mental status score of 8.75 (SD = 2.35). The average CES-D10 score for depressive symptoms was 4.23 (SD = 2.75). Sleep duration showed that 41.73% of participants slept 0–6 h, 49.84% slept 6–8 h, and 8.44% slept more than 8 h. Among participants, 0.35% rated their life as very good, 7.65% as good, 66.10% as average, 23.85% as poor, and 2.06% as very poor. Health behaviors included 42.00% who smoked and 37.81% who drank. Health status revealed that 22.49% had hypertension, 5.06% had diabetes, 6.36% had lung disease, 8.53% had heart disease, 2.77% had liver disease, and 3.84% had kidney disease. Additionally, the one-way analysis of variance revealed statistically significant associations between the number of children and both depressive symptoms and cognitive functioning.

### 3.2. Association between the Number of Children, Depression, and Cognitive Function

We analyzed the relationship between the number of children, depressive symptoms, and cognitive functioning in middle-aged and older adults using the Logarithmic Generalized Linear Models. The results are presented in Table 2.

For episodic memory, in the crude model, participants who had lost a child showed significantly lower situational memory scores compared to participants with one child (*β* = −0.147, 95% CI: −0.280, −0.013, *p* < 0.05). Additionally, participants with two or more children also showed significantly lower situational memory scores (two children: *β* = −0.066, 95% CI: −0.093, −0.040, *p* < 0.001; three children: *β* = −0.125, 95% CI: −0.155, −0.094, *p* < 0.001; four or more children: *β* = −0.222, 95% CI: −0.259, −0.087, *p* < 0.001). As the adjustment variables increased, none of the groups showed statistically significant results except for the group with four or more children (Model 1: *β* = −0.053, 95% CI: −0.091,−0.015, *p* < 0.01; Model 2: *β* = −0.049, 95% CI: −0.087,−0.011, *p* < 0.05). In the analysis of the mental status, the crude model revealed that participants who had lost a child exhibited significantly lower mental status scores (*β* = −0.100, 95% CI: −0.192, −0.009, *p* < 0.05). Additionally, participants with two or more children also showed significantly lower mental status scores (two children: *β* = −0.062, 95% CI: −0.081, −0.043, *p* < 0.001; three children: *β* = −0.091, 95% CI: −0.113, −0.069 *p* < 0.001; four or more children: *β* = −0.125, 95% CI: −0.150, −0.100, *p* < 0.001). As the number of variables increases, the negative correlation between the group with two or more children and mental status becomes statistically significant, except for the lost-child group.

In our analysis of overall cognitive status, we found that in the crude model, participants who had lost a child exhibited significantly lower total cognitive scores (*β* = −0.117, 95% CI: −0.203, −0.031, *p* < 0.01). Additionally, participants with two or more children also exhibited significantly lower total cognitive scores (two children: *β* = −0.064, 95% CI: −0.081, −0.046, *p* < 0.001; three children: *β* = −0.103, 95% CI: −0.123, −0.083 *p* < 0.001; four or more children: *β* = −0.159, 95% CI: −0.183, −0.136, *p* < 0.001). However, the effect of the lost-child and two-children groups was no longer significant in the partially adjusted model (Model 1) and the fully adjusted model (Model 2).

Depressive symptoms were significantly associated with cognitive function across all models. In the crude model, depressive symptoms exhibited a significant negative correlation with situational memory (*β* = −0.141, 95% CI: −0.166, −0.115, *p* < 0.001), mental state (*β* = −0.102, 95% CI: −0.119, −0.085, *p* < 0.001), and the total cognitive score (*β* = −0.116, 95% CI: −0.132, −0.100, *p* < 0.001). The effects of depressive symptoms remained significant in both Model 1 and Model 2 (all *p* < 0.001). In addition, a positive correlation was observed between the lost-child group and the group with two or more children exhibiting depressive symptoms in comparison to the one-child group. Statistically significant results were found across all models (see Appendix A).

Thus, there is a significant association between the number of children and cognitive function, as well as depressive symptoms, in middle-aged and older adults. With increasing covariates, factors influencing cognitive functioning in groups of three children are associated with their mental status. However, cognitive functioning in groups of four or more children is influenced by both episodic memory and mental status. Specifically, participants with three or more children exhibited significantly lower scores across all domains of cognitive function, suggesting that depressive symptoms may play a crucial role in mediating these associations.

### 3.3. Mediating Effect of Depressive Symptoms

To investigate the mediating effect of depressive symptoms between the number of children and cognitive functioning in middle-aged and older adults, we employed the Simultaneous Equation Models (SEMs) and estimated it using three-stage least squares (3SLS). The results of the mediating effects analysis are presented in Table 3. Based on the Akaike Information Criterion (AIC) and the Bayesian Information Criterion (BIC), we identified Model 2 as the most suitable regression model (see Appendix A). We first examined the impact of the number of children on depressive symptoms. The results indicated a significant positive correlation between the number of children and depressive symptoms (the coefficient for path a was 0.019, 95% CI: 0.006, 0.033, *p* < 0.01), suggesting that a higher number of children is associated with more severe depressive symptoms. Next, we assessed the effect of depressive symptoms on cognitive functioning. The findings revealed a significant negative correlation between depressive symptoms and cognitive function (the coefficient for path b was −0.972, 95% CI: −1.159, −0.786, *p* < 0.001), indicating that more severe depressive symptoms are linked to poorer cognitive function. Finally, we analyzed the direct effect of the number of children on cognitive functioning. The results demonstrated a significant negative correlation between the number of children and cognitive functioning (the coefficient for path c was −0.159, 95% CI: −0.248, −0.069, *p* < 0.001), implying that a higher number of children is associated with worse cognitive functioning.

Using 1000 bootstrap tests, we calculated both the indirect and total effects: (1) The total effect (path a + path b + path c) of the number of children on cognitive function was −0.140 (95% CI: −0.228, −0.051, *p* < 0.01). (2) The indirect effect (path a × path b) of the number of children on cognitive functioning through depressive symptoms was −0.019 (95% CI: −0.034, −0.006, *p* < 0.01). (3) The proportion of the mediating effect (pMe) of depressive symptoms between the number of children and cognitive function was 13.57%. Next, the pMe values for each group, based on the number of children, were as follows: no children: 24.31%, two children: 25.39%, three children: 18.45%, and four or more children: 14.74%.

Our findings suggest that depressive symptoms partially mediate the relationship between the number of children and cognitive function in middle-aged and older adults. Specifically, a higher number of children is associated with an increased severity of depressive symptoms, which subsequently leads to a decline in cognitive functioning. This finding underscores the importance of considering mental health factors when examining cognitive function in this demographic. This finding underscores the importance of considering mental health factors when examining cognitive function in middle-aged and older adults.

### 3.4. Sensitivity Analysis

In addition to using Logarithmic Generalized Linear Models and Simultaneous Equation Models (SEMs) with three-stage least squares (3SLS), we also used a Multiple Linear Regression Model to confirm the consistency of the results. The findings indicate that both methods produce essentially the same conclusions, suggesting that the choice of model has a minimal impact on the results (Appendix A).

Next, we divided the data into subsamples based on gender and place of residence (Appendix A). We analyzed the number of children as a continuous variable and also regrouped them (lost, only child, and two or more children) (Appendix A). We found that the main findings remained consistent. Additionally, depression played a more significant mediating role in the relationship between the number of children and cognitive ability in the male group compared to the female group (male: pMe = 17.74%, female: pMe = 11.86%). Furthermore, depression mediated the effect in a much higher proportion of the urban population than in the rural population (urban: pMe = 18.62%, rural: pMe = 7.62%).

With the sensitivity analyses conducted, we confirmed the robustness and reliability of our findings. Whether through alternative modeling, subsample analysis, variable treatment, or bootstrap testing, the results consistently demonstrated a strong relationship between the number of children, depressive symptoms, and cognitive functioning in middle-aged and older adults.

## 4. Discussion

Based on a population-based sample of Chinese individuals, we investigated the association between the number of children and cognitive function in middle-aged and older adults. The results indicate that participants without children, as well as those with multiple children, scored significantly lower in both episodic memory and mental state compared to those with only one child. Both childlessness and having multiple children were identified as risk factors for cognitive decline. For childless individuals, one possible reason for cognitive decline in older adults is their heightened vulnerability to social isolation. Social isolation can indirectly exacerbate cognitive decline by diminishing social engagement and stimulation [40]. Additionally, older adults without children may lack the social and caregiving support typically provided by offspring when care is needed. The absence of both emotional and daily caregiving support may accelerate cognitive deterioration [41]. Families with multiple children may be at an increased risk of cognitive decline for several reasons. First, multiple children’s families often face greater financial pressures. A Chinese study found that additional children can strain family income, potentially impacting the mother’s mental well-being and cognitive abilities [42]. In China, raising multiple children requires a significant financial investment, including education, medical care, and living expenses. These financial pressures can lead to a decline in parents’ quality of life and increase their risk of anxiety and depression. Additionally, financial strain may adversely impact parents’ health behaviors, such as reduced exercise time and irregular work schedules, which can further affect their physical and mental health. In urban areas, the high cost of living—particularly concerning education and housing—intensifies the financial burden on families with several children. Parents may find it necessary to work overtime or hold multiple jobs to cover family expenses, which not only affects their physical health but may also create tensions within the family unit. Furthermore, financial pressure can result in insufficient investment in their children’s education, ultimately hindering their future development. In rural areas, although the cost of living is relatively lower, families with many children still face economic challenges. Farming families rely heavily on agricultural production for their income, and the volatility of agricultural earnings can lead to significant fluctuations in their economic situation. The necessity for families with multiple children to allocate more financial resources from limited means may result in inadequate savings and financial security for parents in their old age, thereby increasing their financial stress and psychological burden. Evidence from northern Europe suggests that higher fertility rates can detract from late-life cognitive function due to diminished economic resources without corresponding gains in social support [43]. Furthermore, economic challenges faced by adult children may negatively impact the memory function of their aging parents [44].

Other contributing factors are culture and the reduction in social support. Traditionally, “more children means more blessings,” but having many children can lead to a “dispersion of responsibilities,” where the presence of multiple siblings dilutes the caregiving responsibilities for aging parents [45]. Studies indicate that as parents age, caregiving responsibilities of parents tend to be more evenly distributed among all children. However, in most cases, families continue to depend primarily on a single child to take on the majority of older people’s caregiving duties [46]. This situation may lead to parents feeling lonely and neglected in their old age, further impacting their mental health and cognitive functioning. In traditional Chinese culture, raising multiple children is viewed as a blessing; however, it also entails significant responsibilities and pressures. In modern society, this perception can create an even greater psychological and emotional burden on parents. First, parents must invest substantial resources in education, healthcare, and living expenses, which can heighten their psychological stress. Parents of large families often struggle to balance their social roles with family responsibilities. This problem is particularly pronounced in rural areas, where parents may need to juggle farm work with childcare, resulting in a dual burden that strains their physical and mental well-being. Moreover, rapid social development and urbanization have significantly altered traditional family structures and social support systems. In contemporary society, the geographical distance between family members has increased, with children often leaving their hometowns for work or education. This separation can prevent parents from receiving timely assistance when they require care. The absence of a robust social support system may exacerbate parents’ feelings of loneliness and helplessness, further compromising their mental health. Influenced by traditional Chinese culture, parents tend to be deeply concerned about their children’s living conditions, even after the children reach adulthood. In comparison to one-child families, the additional burden of parenting, increased geographical distance, and lack of social support in large families have a serious impact on parents’ physical and mental health.

A third factor is the potential limitations on health-related behaviors. Parents of large families often struggle with time management, particularly in maintaining healthy lifestyle habits. They may have insufficient time for exercise [47] or experience sleep deprivation [48]. These limitations on health behaviors increase the risk of chronic conditions, such as diabetes and hypertension, which are closely linked to cognitive decline [49,50,51]. Collectively, these factors contribute to the heightened likelihood of cognitive decline in individuals with multiple children. Caring for multiple children can make it challenging for parents to find time to exercise or maintain a healthy lifestyle. As a result, parents may neglect their own health needs, such as regular checkups and health management, while focusing on their children’s care. This neglect can lead to a buildup of health issues that adversely affect their cognitive functioning and overall health. Additionally, parents in families with several children may face challenges in their education and career development. The demands of caring for multiple children may prevent them from working full-time or pursuing advanced career opportunities, which can result in limited professional growth and lower income levels. This restricted career development not only impacts the family’s financial situation but can also lead to psychological frustration and stress for the parents, further compromising their mental health. In terms of education, parents of large families may struggle to provide equal educational resources and opportunities for each child, potentially resulting in significant disparities in educational attainment among siblings. This uneven distribution of educational resources can create tensions within the family, further affecting the parents’ mental health and overall family harmony.

In analyzing overall cognitive function, this study adjusted the models to account for potential confounding variables. In the unadjusted model, participants without children had significantly lower total cognitive scores compared to those with one child. However, after adjusting for confounders, this effect was no longer significant, suggesting that the influence of being childless on cognitive function may be mitigated by factors such as marital status or external social support [52]. In contrast, even after adjusting for confounders, participants with three or more children continued to show significantly lower cognitive scores, indicating that having three or more children may place individuals at a cognitive disadvantage. This finding is consistent with previous research [43].

This study found a significant negative correlation between depressive symptoms and cognitive function in middle-aged and older Chinese adults. More severe depressive symptoms were linked to lower episodic memory and mental state scores, consistent with previous findings [53,54]. These results suggest that depressive symptoms may mediate the link between the number of children and cognitive function by influencing psychological and global cognitive scores. Further analysis confirmed that mediation. In our study, a significant positive correlation was found between having more children and worsening depressive symptoms, which in turn were linked to cognitive decline [53,55]. Specifically, participants from families without children exhibited more severe depressive symptoms (Appendix A), leading to a notable decline in their cognitive abilities. Notably, the proportion of mediating effects attributed to depressive symptoms was greater in families without children and in families with two children, underscoring the importance of depressive symptoms in elucidating the decline in cognitive functioning within these families.

Our study also found that the mediating effect of depressive symptoms on cognitive function was more obvious in male participants and urban residents. This may be attributed to the fact that in traditional Chinese society, men often bear greater economic and familial responsibilities. The increased burden from these responsibilities may exacerbate depressive symptoms, which in turn can negatively impact cognitive function. Research has shown that economic pressure tends to have a more pronounced negative effect on men’s mental health compared to women, likely due to the higher societal expectations and role demands placed on men during times of financial strain [56]. Additionally, men of lower socioeconomic status are more prone to cognitive impairments or dementia, suggesting that low income and economic instability may have a more detrimental impact on men’s cognitive function [57]. Additionally, studies have shown that urban residents face a higher risk of mental health issues compared to their rural counterparts, with depression and anxiety being the most common mental health concerns associated with urbanization [58,59]. Urbanization often leads to weakened social ties, which can exacerbate depressive symptoms in older adults [60]. These findings suggest that the influence of depressive symptoms on cognitive function can significantly vary based on gender and residential status, with increased depressive symptoms directly or indirectly impairing cognitive function.

However, we acknowledge that this study has several limitations. First, we combined data from the cohort studies of waves 1–5 for sample processing and excluded, to the extent possible, individuals with pre-existing psychiatric disorders in the baseline population, as well as factors that might influence the results of the analyses, while considering the issue of recall bias concerning covariates. The analyses of recall bias, correlation, and mediation effects were conducted using cross-sectional data from wave 5 (2020), which may not provide as robust evidence for causality as the cohort studies. Second, the CES-D10 questionnaire was administered through self-reports from the sample population, which may introduce a risk of recall bias and inaccurate responses. The CES-D10 is a widely utilized tool for assessing clinically significant depressive symptoms; however, it serves only as a screening instrument and is not intended for diagnosing depression [61]. In the assessment of cognitive functioning, some results were derived from telephone interviews, which may have introduced bias into the findings. Furthermore, although we utilized the number of living children to mitigate a potential recall bias, the covariates related to smoking, alcohol consumption, sleep duration, and life satisfaction were also obtained through self-reports from respondents. The imputation of missing values was based on the survey data from wave 5 (2020) and interpolation, which may have influenced the analysis of the results. This study is based on conclusions drawn from the Chinese middle-aged and older population, and the generalizability of these conclusions may be limited due to variations in economic conditions and cultural practices across different countries. The findings of this study may be subject to certain limitations attributable to the distinctive demographic and cultural context of China. It is essential to consider China’s unique socioeconomic environment and cultural factors when interpreting these results. Furthermore, while the current study offers valuable insights into the relationship between the number of children, cognitive functioning, and depressive symptoms, these findings may not be entirely generalizable to other countries or cultural contexts. Consequently, it is imperative to thoroughly account for these cultural and demographic characteristics when formulating relevant policies and interventions. Future study directions that focus on the long-term effects of family structure on mental health and cognitive function in different cultures.

## 5. Conclusions

This study examined the associations between the number of children, depressive symptoms, and cognitive functioning in middle-aged and older adults. Both childlessness and having multiple children were identified as risk factors for cognitive decline. Depressive symptoms partially mediated the association between the number of children and cognitive functioning. Specifically, participants with multiple children demonstrated a significantly heightened risk of cognition decline, even after controlling for covariates. The proportion of the mediating effect attributed to depression was more pronounced in middle-aged and older adults who were without children or had two or more children. Moreover, the mediating effect was found to be more significant among men and urban residents. These results underscore the importance of considering mental health factors when studying cognitive function in middle-aged and older populations.

## Figures and Tables

**Figure 1 healthcare-12-01928-f001:**
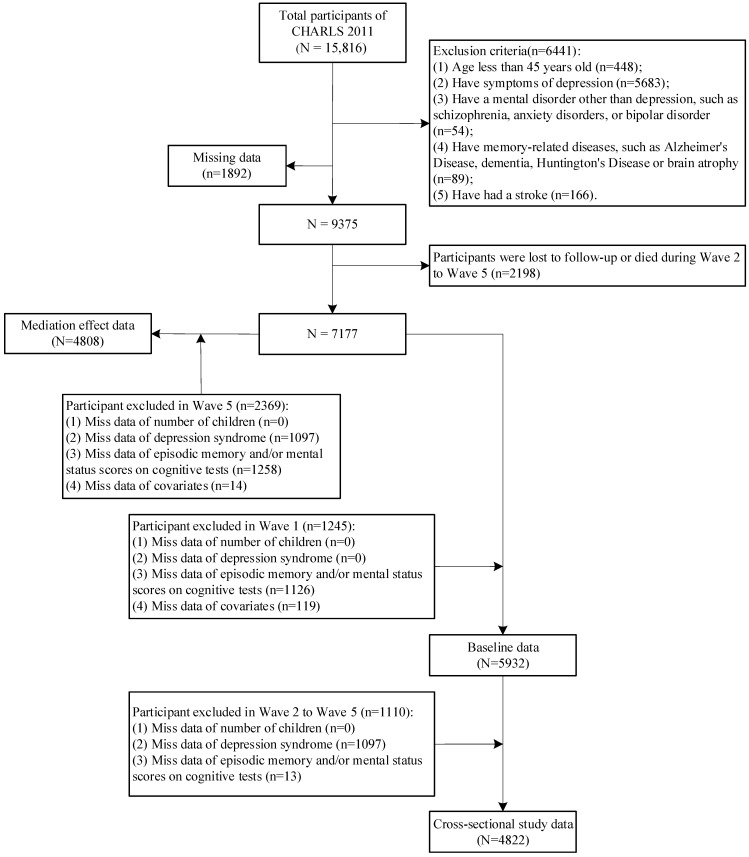
The flowchart of this study design.

**Table 1 healthcare-12-01928-t001:** Baseline characteristics of participants.

Variables, n (%)	Number of Children (N = 5932)					χ^2^/t-Statistic	*p*-Value
	0 (n = 55)	1 (n = 1102)	2 (n = 2268)	3 (n = 1386)	≥4 (n = 1121)		
**Age** (mean ± SD, years)	56.91 ± 7.85	53.13 ± 6.35	54.73 ± 6.92	58.40 ± 7.91	64.07 ± 8.19	1500.000	<0.001 ***
**Gender**							
Female	16 (0.58)	527 (19.21)	1035 (37.73)	625 (22.79)	540 (19.69)	10.365	0.035 *
Male	39 (1.22)	575 (18.03)	1233 (38.66)	761 (23.86)	581 (18.22)		
**Residence**							
Rural	29 (0.85)	366 (10.75)	1323 (38.84)	910 (26.72)	778 (22.84)	369.681	<0.001 ***
Urban	26 (1.03)	736 (29.14)	945 (37.41)	476 (18.84)	343 (13.58)		
**Education level**							
Below Primary School	24 (1.21)	201 (10.11)	663 (33.33)	520 (26.14)	581 (29.21)	507.464	<0.001 ***
Primary School	11 (0.81)	199 (14.62)	506 (37.18)	365 (26.82)	280 (20.57)		
Middle School	12 (0.76)	375 (23.64)	699 (44.07)	338 (21.31)	162 (10.21)		
High School and Above	8 (0.80)	327 (32.83)	400 (40.16)	163 (16.37)	98 (9.84)		
**Marital status**							
Divorced/Separated/Widowed/Never Married	21 (4.95)	66 (15.57)	99 (23.35)	82 (19.34)	156 (36.79)	189.050	<0.001 ***
Married/Cohabitated	34 (0.62)	1036 (18.81)	2169 (39.38)	1304 (23.67)	965 (17.52)		
**Total Cognition** (mean ± SD, score)	12.16 ± 3.34	13.80 ± 2.99	13.01 ± 3.14	12.37 ± 3.24	11.48 ± 3.51	502.032	<0.001 ***
Episodic Memory	3.58 ± 1.54	4.42 ± 1.63	4.11 ± 1.62	3.79 ± 1.60	3.45 ± 1.64	318.144	<0.001 ***
Mental Status	8.58 ± 2.42	9.39 ± 1.96	8.90 ± 2.22	8.58 ± 2.42	8.02 ± 2.64	246.147	<0.001 ***
**CES-D10** ^a^ (mean ± SD, score)	4.70 ± 2.30	3.79 ± 2.64	4.15 ± 2.75	4.46 ± 2.80	4.51 ± 2.76	101.993	<0.001 ***
**Sleep Duration(h)**							
(0, 6]	17 (0.69)	404 (16.32)	899 (36.32)	628 (25.37)	527 (21.29)	52.395	<0.001 ***
(6, 8]	34 (1.15)	617 (20.87)	1178 (39.85)	644 (21.79)	483 (16.34)		
>8	4 (0.80)	81 (16.17)	191 (38.12)	114 (22.75)	111 (22.16)		
**Life Satisfaction**							
Very Good	1 (4.76)	3 (14.29)	5 (23.81)	3 (14.29)	9 (42.86)	50.770	<0.001 ***
Good	7 (1.54)	88 (19.38)	153 (33.70)	105 (23.13)	101 (22.25)		
Fair	35 (0.89)	752 (19.18)	1562 (39.85)	903 (23.04)	668 (17.04)		
Poor	10 (0.71)	231 (16.33)	508 (35.90)	345 (24.38)	321 (22.69)		
Very poor	2 (1.64)	28 (22.95)	40 (32.79)	30 (24.59)	22 (18.03)		
**Smoking**	29 (1.16)	434 (17.43)	961 (38.59)	591 (23.73)	475 (19.08)	6.123	0.190
**Drinking**	23 (1.03)	478 (21.31)	897 (39.99)	501 (22.34)	344 (15.34)	43.632	<0.001 ***
**Hypertension**	12 (0.90)	237 (17.77)	453 (33.96)	337 (25.26)	295 (22.11)	20.925	<0.001 ***
**Diabetes**	1 (0.33)	69 (23.00)	105 (35.00)	76 (25.33)	49 (16.33)	7.016	0.135
**Lung Disease**	8 (2.12)	47 (12.47)	134 (35.54)	88 (23.34)	100 (26.53)	27.447	<0.001 ***
**Stroke** ^b^	0 (0)	0 (0)	0 (0)	0 (0)	0 (0)	-	-
**Mental Disorder** ^c^	0 (0)	0 (0)	0 (0)	0 (0)	0 (0)	-	-
**Heart Disease**	7 (1.38)	79 (15.61)	175 (34.58)	138 (27.27)	107 (21.15)	10.881	0.028 *
**Liver Disease**	4 (2.44)	41 (25.00)	51 (31.10)	36 (21.95)	32 (19.51)	10.327	0.035 *
**Kidney Disease**	7 (3.07)	39 (17.11)	90 (39.47)	56 (24.56)	36 (15.79)	13.474	0.009 **

a, b, c: For research design purposes, participants with baseline depression (CES-D10 score ≥ 10), a history of stroke, or mental disorders affecting cognitive function were excluded from the study. * *p* < 0.05, ** *p* < 0.01, *** *p* < 0.001.

**Table 2 healthcare-12-01928-t002:** Associations between number of children, depression, and cognitive function in older adults based on cross-sectional study data.

Models	Episodic Memory		Mental Status		Total Cognition	
	*β*	95% CI	*β*	95% CI	*β*	95% CI
**Crude model**						
**Number of Children**						
1	Ref.	-	Ref.	-	Ref.	-
0	−0.147 *	(−0.280, −0.013)	−0.100 *	(−0.192, −0.009)	−0.117 **	(−0.203, −0.031)
2	−0.066 ***	(−0.093, −0.040)	−0.062 ***	(−0.081, −0.043)	−0.064 ***	(−0.081, −0.046)
3	−0.125 ***	(−0.155, −0.094)	−0.091 ***	(−0.113, −0.069)	−0.103 ***	(−0.123, −0.083)
≥4	−0.222 ***	(−0.259, −0.185)	−0.125 ***	(−0.150, −0.100)	−0.159 ***	(−0.183, −0.136)
**Depression**						
0	Ref.	-	Ref.	-	Ref.	-
1	−0.141 ***	(−0.166,−0.115)	−0.102 ***	(−0.119,−0.085)	−0.116 ***	(−0.132, −0.100)
**Model 1**						
**Number of Children**						
1	Ref.	-	Ref.	-	Ref.	-
0	−0.055	(−0.178, 0.068)	−0.053	(−0.137, 0.031)	−0.054	(−0.131, 0.004)
2	−0.007	(−0.033, 0.018)	−0.020 *	(−0.038, −0.002)	−0.015	(−0.031, 0.002)
3	−0.018	(−0.049, 0.013)	−0.031 **	(−0.053, −0.010)	−0.026 **	(−0.045, −0.006)
≥4	−0.053 **	(−0.091, −0.015)	−0.046 ***	(−0.071, −0.020)	−0.048 ***	(−0.071, −0.024)
**Depression**						
0	Ref.	-	Ref.	-	Ref.	-
1	−0.106 ***	(−0.130, −0.083)	−0.062 ***	(−0.078, −0.046)	−0.078 ***	(−0.093, −0.063)
**Model 2**						
**Number of Children**						
1	Ref.	-	Ref.	-	Ref.	-
0	−0.061	(−0.183, 0.061)	−0.060	(−0.143, 0.024)	−0.060	(−0.137, 0.016)
2	−0.006	(−0.032, 0.019)	−0.018 *	(−0.036, −0.001)	−0.013	(−0.029, 0.003)
3	−0.015	(−0.046, 0.015)	−0.028 *	(−0.049, −0.007)	−0.023 *	(−0.042, −0.003)
≥4	−0.049 *	(−0.087, −0.011)	−0.041 ***	(−0.067, −0.016)	−0.043 ***	(−0.067, −0.020)
**Depression**						
0	Ref.	-	Ref.	-	Ref.	-
1	−0.102 ***	(−0.127, −0.077)	−0.063 ***	(−0.080, −0.046)	−0.078 ***	(−0.093, −0.062)

Crude model: No adjustments. Model 1: Partially adjusted model, adjusting for age, sex, residence, education level, and marital status. Model 2: Fully adjusted model, additionally adjusting for smoking, drinking, life satisfaction, sleep duration, hypertension, diabetes, lung disease, stroke, mental disorder, heart disease, liver disease, and kidney disease. Ref: One child was chosen as the reference value, considering that the participants were affected by the one-child policy in China. * *p <* 0.05, ** *p* < 0.01, *** *p <* 0.001.

**Table 3 healthcare-12-01928-t003:** Relationship between the number of children and cognitive function mediated by depressive symptoms.

	Models	Crude Model	Model 1	Model 2
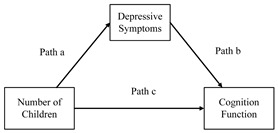 Path a: Exposure predicting mediator Path b: Mediator predicting outcome Path c: Exposure predicting outcome	**Coefficients** **(95% CI)**			
	Path a	0.035 *** (0.022, 0.047)	0.020 ** (0.006, 0.034)	0.019 ** (0.006, 0.033)
	Path b	−1.336 *** (−1.531, −1.142)	−0.955 *** (−1.135, −0.773)	−0.972 *** (−1.159, −0.786)
	Path c	−0.613 *** (−0.701, −0.524)	−0.171 *** (−0.261, −0.081)	−0.159 *** (−0.248, −0.069)
	**Effect**			
	Total effect ^a^	−0.566 *** (−0.653, −0.479)	−0.152 *** (−0.241, −0.063)	−0.140 ** (−0.228, −0.051)
	Indirect effect ^b^	−0.046 *** (−0.065, −0.029)	−0.019 ** (−0.034, −0.005)	−0.019 ** (−0.034, −0.006)
	pMe ^c^	8.13%	12.50%	13.57%

Crude model: No adjustments. Model 1: Partially adjusted model, adjusting for age, sex, residence, education level, and marital status. Model 2: Fully adjusted model, additionally adjusting for smoking, drinking, life satisfaction, sleep duration, hypertension, diabetes, lung disease, stroke, mental disorder, heart disease, liver disease, and kidney disease. a: Total effects represent the overall impact of the number of children and depression on cognitive functioning, expressed through regression coefficients with 95% confidence intervals (CIs). b: Indirect effect = Multiplication of path a and path b coefficients. Coefficients for indirect effects were determined using the bootstrap method with 1000 repetitions, and a 95% bias-corrected and accelerated (BCa) confidence interval (CI) was employed to enhance the accuracy of the analysis. c: pMe is the proportion of the total effect that is mediated. * *p* < 0.05, ** *p* < 0.01, *** *p* < 0.001.

## Data Availability

Data available in a publicly accessible repository. The data from the 2011, 2013, 2015, 2018, and 2020 China Health and Retirement Longitudinal Study (CHARLS) are publicly available at https://charls.charlsdata.com/pages/data/111/en.html, accessed on 16 November 2023.

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
