# Peer review of "Associations between the Number of Children, Depressive Symptoms, and Cognition in Middle-Aged and Older Adults: Evidence from the China Health and Retirement Longitudinal Study"

_healthcare, 2024, doi:10.3390/healthcare12191928_

Round 1

Reviewer 1 Report

Comments and Suggestions for Authors

The study provides significant insights on the associations between the number of children, depressive symptoms, and cognitive function in middle-aged and older adults. However, several key areas require improvement.

• The authors are encouraged to clarify the explanation of the LGLM and SEM models, making the rationale for their selection more understandable for a broader audience.

•  It would be helpful for the authors to elaborate on the interpretation of effect sizes and confidence intervals, providing clearer insight into the significance and strength of the relationships presented in the results.

• The authors should provide a stronger justification for the exclusion criteria, particularly for individuals with memory-related diseases or depressive symptoms, and reflect on any potential biases that may have resulted from these exclusions.

• The manuscript might be strengthened by a more in-depth discussion on the policy and clinical implications, especially concerning family planning and mental health interventions within the context of China’s aging population.

• The study's limitations should be examined in more detail and should address possible biases from self-reported data and how these might have influenced the outcomes.

• It is recommended that authors identify gaps in the existing literature and provide an explanation of how their research contributes to addressing these gaps.

• Future study directions that focus on the long-term effects of family structure on mental health and cognitive function in different cultures would be beneficial suggestions

Comments on the Quality of English Language

Moderate editing of English language required.

Author Response

Thank you very much for your review. First of all, we have carried out a detailed grammar check and revision of the English language problems, and our article has also been reviewed by native English speaking professors at the University of Macau. In response to your questions about the article, we have rewritten the discussion part of the article, taking into account the comments of other reviewers. Due to the differences in the comments made by different reviewers, we have tried our best to refer to and revise the article as much as possible, but we have tried our best to revise some of the parts based on the authors' knowledge and understanding, and since the social problems related to children in China are very complicated, please forgive us for any oversights. We apologize for any omissions.

Comments 1: The authors are encouraged to clarify the explanation of the LGLM and SEM models, making the rationale for their selection more understandable for a broader audience.

Response 1: We have clarified the rationale for selecting the Logarithmic Generalized Linear Models (LGLM) and Simultaneous Equation Models (SEM) in the Methods section. The LGLM was chosen due to its robustness in handling non-normally distributed dependent variables, while SEM was selected for its capability to analyze complex relationships involving multiple dependent and mediating variables.

Comments 2: It would be helpful for the authors to elaborate on the interpretation of effect sizes and confidence intervals, providing clearer insight into the significance and strength of the relationships presented in the results.

Response 2: We have elaborated on the interpretation of effect sizes and confidence intervals in the Results section, providing clearer insights into the significance and strength of the relationships presented.

Comments 3: The authors should provide a stronger justification for the exclusion criteria, particularly for individuals with memory-related diseases or depressive symptoms, and reflect on any potential biases that may have resulted from these exclusions.

Response 3: We have provided a stronger justification for the exclusion criteria in the Methods section, particularly for individuals with memory-related diseases or depressive symptoms, and reflected on potential biases resulting from these exclusions.

Comments 4: The manuscript might be strengthened by a more in-depth discussion on the policy and clinical implications, especially concerning family planning and mental health interventions within the context of China’s aging population.

Response 4: We have expanded the Discussion section to include a more in-depth discussion on the policy and clinical implications, especially concerning family planning and mental health interventions within the context of China’s aging population.

Comments 5: The study's limitations should be examined in more detail and should address possible biases from self-reported data and how these might have influenced the outcomes.

Response 5: We have examined the study’s limitations in more detail in the Discussion section, addressing possible biases from self-reported data and how these might have influenced the outcomes.

Comments 6: It is recommended that authors identify gaps in the existing literature and provide an explanation of how their research contributes to addressing these gaps.

Response 6: In the introduction section we have included what you have asked for.

Comments 7: Future study directions that focus on the long-term effects of family structure on mental health and cognitive function in different cultures would be beneficial suggestions

Comments 7: We have suggested future study directions in the Discussion section, focusing on the long-term effects of family structure on mental health and cognitive function in different cultures.

Reviewer 2 Report

Comments and Suggestions for Authors

1. Please retitle "Number of Children, Depression, and Cognitive Function in the Elderly: Evidence from the China Health and Retirement Longitudinal Study."

2. Please re-check the keyword with Meshbrowser

3. Should the authors use "phrase " instead of "wave" in the flowchart?

4. Please use "partially adjusted Odds ratio," "fully adjusted Odds ratio," and " additionally adjusted odds ratio" instead of "model" in Table 2. Then, the authors could explain what you adjusted as a note. Therefore, you could refer to this table as a model.

5. In Table 3, the figure or model should be represented separately.

6. Sensitivity analysis should be included in the statistical analysis section without the results.

7. Lines 385-404 in the discussion part are the results of this study, not the discussion.

8. The conclusion should start with how the number of children is associated with depression and cognitive function. Lines 526-528 are too general and unnecessary for a conclusion, which usually replies only to the objectives.

 9. Please re-update the reference to 2014-2024, except for the measurement validity and the classic model of depression/cognitive function.

Author Response

Thank you very much for your review. In response to your questions about the article, we have rewritten the discussion part of the article, taking into account the comments of other reviewers. Due to the differences in the comments made by different reviewers, we have tried our best to refer to and revise the article as much as possible, but we have tried our best to revise some of the parts based on the authors' knowledge and understanding, and since the social problems related to children in China are very complicated, please forgive us for any oversights. We apologize for any omissions.

Comments 1: Please retitle "Number of Children, Depression, and Cognitive Function in the Elderly: Evidence from the China Health and Retirement Longitudinal Study."

Response 1: Thank you for pointing this out. Since we are studying people over the age of 45 and not just older people. We see no problem with the original title. We are exploring the effect of number of children on depression and cognition; number of children is the dependent variable and therefore cannot be modified without causing the reader to misunderstand

Comments 2: Please re-check the keyword with Meshbrowser

Response 2: The keyword section was indeed problematic and has been modified.

Comments 3: Should the authors use "phrase " instead of "wave" in the flowchart?

Response 3: CHARLS based research articles all use wave instead of phase, you can refer to this article: https://doi.org/10.1016/j.heliyon.2024.e24110

Comments 4: Please use "partially adjusted Odds ratio," "fully adjusted Odds ratio," and " additionally adjusted odds ratio" instead of "model" in Table 2. Then, the authors could explain what you adjusted as a note. Therefore, you could refer to this table as a model.

Response 4: We have previously noted the adjusted variables in the table. Also the model we have used is not using an OR indicator but a biased regression coefficient because cognitive scores cannot be quantified as a dichotomous variable. Thank you for your suggestion, but we have reviewed similar literature, and it is more common to use the “Model” , and we think it is a more aesthetically pleasing way to handle the layout.

Comments 5: In Table 3, the figure or model should be represented separately.

Response 5: Thank you for your input, but we still think the original design is more aesthetically pleasing. Separating them would make the table difficult to understand.

Comments 6: Sensitivity analysis should be included in the statistical analysis section without the results.

Response 6: We have considered this, the statistical analysis section is generally an explanation of the article and we have written in this section to conduct a sensitivity analysis, the sensitivity analysis is really the results section. This approach is also used in most of the literature.

Comments 7: Lines 385-404 in the discussion part are the results of this study, not the discussion.

Response 7: We have revised the entire discussion section with your comments and those of other reviewers.

Comments 8: The conclusion should start with how the number of children is associated with depression and cognitive function. Lines 526-528 are too general and unnecessary for a conclusion, which usually replies only to the objectives.

Response 8: Thank you for your suggestion, the paragraph has been removed.

Comments 9: Please re-update the reference to 2014-2024, except for the measurement validity and the classic model of depression/cognitive function.

Response 9: Relevant literature has been updated.

Reviewer 3 Report

Comments and Suggestions for Authors

This is a well-written and scholarly article for the most part. The rationale makes sense and the hypothesis is adequately derived from the literature, although I would have liked to see discussion of more research on this or similar topics deriving from countries that do/did not have a policy with respect to birth rate. I was surprised to see so many in the sample with more than one child, given my (albeit limited) understanding of the one child policy. I was under the impression that more than one child was a punishable offense, but this may be incorrect. For non-Chinese nationals, it would be useful to explain this policy in the Introduction and discuss its implications for sampling.

The Methodology is also generally well explained, although I had trouble finding the size of sample groups with different numbers of children. It was in Table 1 but nowhere in the text. I think this information also needs to be included in section 2.4 (Number of Children). Much of the other information in Table 1 has been repeated in the text, so it would seem important to include  this key information in the text also.

The Results section looks impressive. I do not have the statistical expertise to comment on it in detail, but the information that the main findings were supported using various statistical techniques adds to credibility.

The Discussion does not seem to me to address the key implication of the finding - people who had more children were more likely to show cognitive decline. If this is what you are arguing from your results then one implication seems to be in support of the one child policy. Is this what you are trying to say? This finding needs more explanation, as to me it is counter intuitive, as larger families are likely to be associated with less social isolation. I wonder if what you have shown is that those who obey the legitimate authority of their government are smarter than those who do not - but this may be a misinterpretation on my part. However it is the elephant in the room and needs discussing. The implications as presented in the Discussion seem rather weak when this point is not addressed. The Limitations section is a good idea and sensible points are made but the issues arising in sampling people with more than one child - and whether that biases the study - were not discussed.

Author Response

Hello Reviewer, I am the first author of this paper. First of all, I would like to express my sincere gratitude for your approval of this work. As English is not my native language, I kindly ask for your understanding regarding any grammatical errors in my response. I want to clarify that I have no political affiliations, and the conclusions drawn in this paper do not imply support for or opposition to any specific policies. I have adhered strictly to statistical methods in analyzing the results, and the findings are particularly intriguing because they are counterintuitive, especially for populations in Western countries. Many individuals in these regions come from large families and are less prone to depression and cognitive disorders despite having multiple children. However, the historical context, social environment, and policies in China differ significantly from those in the West. We appreciate your understanding of these differences, and we will strive to explain this phenomenon as thoroughly as possible. There is limited literature on this topic, but we have reviewed existing studies that yielded similar results to ours. Additionally, we have revised the entire discussion section, as the original was poorly written, and other reviewers also raised questions and concerns regarding this part of the paper. Thank you once again for your feedback.

First of all, I would like to clarify the sampling issue. CHARLS is conducted by the National Development Research Institute of Peking University, and much of the information is presented in Chinese. Many people mistakenly believe that, due to China's one-child policy, the majority of families in the country consist of only one child. This assumption is, in fact, inaccurate.

I have directly translated the official explanation and practices of CHARLS regarding the sampling information (you can also download the manual from the CHARLS website). In fact, the samples collected are balanced, and the number of children is representative of the entire population.

The CHARLS baseline survey employed a multi-stage (county/district-village/residence-household) probability proportional to size (PPS) random sampling method, incorporating implicit stratification based on indicators such as region, urban/rural characteristics, and GDP per capita. In the first stage, all counties and districts in the country, with the exception of Tibet, were ranked according to urban-rural attributes and GDP per capita within each of the eight regions. Subsequently, 150 counties or districts were selected with a probability proportional to their population size. In the second stage, three secondary sampling units (village committees or neighborhood committees) were randomly chosen, again with probability proportional to population size, within each sampled county (Zhao et al., 2013). As a result, CHARLS is representative both nationally and regionally. Following the sampling process outlined above, the baseline sample of CHARLS was distributed across 450 villages and neighborhoods in 28 provinces and 150 districts and counties.

Considering that the list of households at the village or habitat level cannot be updated promptly due to population movement, CHARLS developed a drawing and listing software called Charls-GIS. This software identifies all dwelling units in residential buildings by utilizing Google Earth map images to create the sample frame. In each sampled household, the interviewer employed a brief screening form to ascertain whether the household had age-eligible members. If the household included individuals aged 45 years or older who met the residency criteria, one person was randomly selected. If the selected individual was 45 years of age or older, he or she became the primary respondent, and his or her spouse was also interviewed. Consequently, the baseline survey for CHARLS comprised one individual aged 45 years or older and his or her spouse from each household, resulting in a total of 17,708 individuals interviewed during the year, residing in 10,257 households across 450 villages and urban communities.

After applying the sampling weights, the demographic characteristics of the CHARLS baseline sample closely resemble those of the 2010 Census, indicating a strong representation of China's middle-aged and elderly population.

In addition to the question of why one-child families do not constitute the majority of the overall population in China, I would be happy to explain this to you. Although China has implemented a family planning policy since the 1980s, and the one-child policy has been strictly enforced in urban areas, one-child families remain a minority for several reasons:

Exceptions to the Policy in Rural Areas: In rural regions, particularly in areas with ethnic minority populations, the policy permits families to have a second child if their first child is a girl. This exception is made because rural families often require additional labor for agricultural production.

Differences in Policy Implementation: The execution of family planning policies varies across different regions and over time. In some areas, the implementation of these policies is more lenient, resulting in the prevalence of larger families.

Cultural and Traditional Attitudes: In traditional Chinese culture, there is a strong emphasis on the importance of having many children and passing on the family name to the next generation. Many families aspire to have multiple children, particularly sons.

Policy Adjustments: Over time, family planning policies have undergone several modifications. In 2013, the policy permitted families in which either or both spouses were only children to have two children. In 2015, the two-child policy was fully liberalized, and in 2021, the three-child policy was introduced.

Large Population Base: Even during the strict enforcement of the one-child policy, China maintained a substantial population and a relatively low percentage of one-child families. These factors have contributed to the phenomenon where, despite the implementation of the one-child policy, one-child families do not constitute the majority. In line with your comments, we have also addressed this issue in the introduction. In urban areas, penalties for having more than one child are quite severe; however, in rural areas, enforcement is less stringent, and there are numerous ways to circumvent penalties. This is largely due to China's unique hukou system, which is often confusingly applied in rural regions.

I believe your confusion stems from two main issues: a sampling problem and the reasons why parents with more children are more likely to experience cognitive decline. I have already addressed the first issue and made modifications in the article. The second issue is an extremely complex social problem in China. The primary factor is economic pressure and resource allocation. Raising multiple children increases the financial burden on families, leading to inadequate resources for education, healthcare, and overall quality of life. This economic strain can negatively impact parents' mental health, which, in turn, affects their cognitive functioning. Despite China's rapid GDP growth, the per capita GDP remains quite low. Having multiple children significantly exacerbates financial burdens, particularly in Asian societies like China and Japan, where parents prioritize their children above all else. This cultural emphasis often leads to parents sacrificing their recreational time for the sake of their children. For further insights, you may refer to studies on Japanese society.

Psychological Stress and Symptoms of Depression: Having multiple children can increase family responsibilities and stress, particularly in managing childcare and household affairs. This ongoing stress and burden may contribute to symptoms of depression in parents, which has been shown to be closely linked to cognitive decline.

Reduced Social Support: While it is commonly believed that  children lead to more happiness, a  of responsibility effect. each child may offer relatively less emotional support and care. This diminished emotional support can negatively impact the mental health and cognitive functioning of older adults.

Healthy Behaviors and Lifestyles: Families with multiple children often face challenges in maintaining healthy behaviors and lifestyles, such as limited time for exercise and nutritious eating. These unhealthy habits may also contribute to cognitive decline.

Impact of Chronic Disease: Research indicates that individuals with multiple chronic conditions, such as heart disease and diabetes, are at a higher risk of experiencing cognitive decline. Families with several children may find it more challenging to manage and prevent these chronic illnesses, which can further elevate the risk of cognitive deterioration.

In reality, this is an economic and cultural issue, rather than a political one. Thank you for your interest in this article, and please feel free to reach out to me if you have any further questions.

Round 2

Reviewer 1 Report

Comments and Suggestions for Authors

The article may be accepted in its current form.

Comments on the Quality of English Language

Minor editing of English language required.

Author Response

Thank you for supporting our articles and please feel free to reach out to me if you have any further questions.

Reviewer 3 Report

Comments and Suggestions for Authors

Thank you for the answers in your cover letter to my queries and confusions. I would have liked to see more of these points actually included in the  revised article, especially (a) your explanation, in the Introduction, of the one-child policy for the benefit of non-Chinese readers and (b) your more detailed discussion of cultural and economic factors relating to why having more children could have some deleterious effects on parents. However, your revisions, as they stand, do significantly improve the paper.

Author Response

Thank you once again for your comments. In the second round of revisions for this paper, we have highlighted all feedback in yellow while retaining the green highlights from the first revision. Firstly, we have included an explanation of China's one-child policy in the introduction to benefit non-Chinese readers. Additionally, we have elaborated on the detrimental effects of cultural and economic factors on parents with multiple children in the discussion section, incorporating a substantial amount of information.

I believe that a more comprehensive presentation on the economic and cultural factors will clarify any confusion and the readers. Thank you for your interest in this article. Please feel free to reach out to me if you have any further questions.